

# Earth system models overestimate the sensitivity of apparent oxygen utilisation to age change in the deep ocean

*Damien Couespel[a], Xabier Davila[a], Nadine Goris[a], Emil Jeansson[a], Siv K. Lauvset[a], Jerry Tjiputra[a]

[a]NORCE Research AS, Bjerknes Centre for Climate Research, Bergen, Norway

Corresponding author: daco@norceresearch.no

## Abstract

The biological carbon pump (BCP), involving photosynthesis at the surface and remineralisation at depth, maintains a significant vertical gradient in dissolved inorganic carbon (DIC), promoting the ocean's ability to absorb atmospheric $CO_2$. Remineralised DIC is a good indicator of the strength of the BCP. It can be estimated from apparent oxygen utilisation (AOU) that measures the deficit of oxygen compared to saturation. AOU is projected to increase under climate change due to changes in remineralisation rates and circulation. However, the amplitude of the change is still uncertain. Here, we identify linear relationships between AOU trends and age trends in the deep ocean in simulations of the contemporary (1972-2013) and future (2015-2099) periods from five Earth system models (ESMs). Linear relationships identified within observational data for the contemporary period indicate that ESMs overestimate the sensitivity of AOU to age changes in the deep ocean. The study highlights the stability over time of the AOU sensitivity to age changes, suggesting an overestimation of the BCP strengthening inferred from AOU. Furthermore, our analysis underscores the substantial role of circulation slowdown in increasing remineralised DIC. These insights emphasise the challenges and opportunities to constrain future BCP projections due to circulation uncertainties.



## Introduction

The capacity of the ocean to take up and store carbon is driven by the marine carbonate chemistry, the solubility pump and the biological carbon pump (BCP hereafter, accounting for the carbonate and soft-tissue pumps, Volk and Hoffert [1985] and see DeVries [2022] for a review of the ocean carbon cycle). A part of the BCP is the photosynthetic transformation of inorganic carbon to organic carbon at the surface. The organic material is then transported to depth where it is transformed back into its inorganic form through remineralization. In the deep ocean, remineralised carbon and nutrients are accumulated. This accumulation is an important component of the BCP and is connected to the strength of the ocean circulation. Then, inorganic carbon and nutrients are transported back to the surface, closing the loop. The BCP is thought to be responsible for keeping the concentration of dissolved inorganic carbon (DIC) at the surface low, resulting in a large vertical gradient of DIC [Volk and Hoffert, 1985, Boyd et al., 2019, DeVries, 2022] with more DIC in the deep ocean. This enhances the capacity of the ocean to take up atmospheric $CO_2$ [Kwon et al., 2009]. Without the BCP, atmospheric $CO_2$ would be more than 200 ppm higher [Sarmiento and Toggweiler, 1984, Maier-Reimer et al., 1996, Goodwin et al., 2008, Tjiputra et al., 2025].

Due to competition between the decrease in organic matter export and slower circulation, it is unclear how the role of BCP will change in the future [Frenger et al., 2024]. There is a general consensus between state-of-the-art Earth system models (ESMs) that the BCP and the processes involved are impacted by global warming [Wilson et al., 2022], but the amplitude of the change and its response to higher atmospheric $CO_2$ are both still uncertain. Indicators of the functionality of the BCP are primary production (related to the photosynthetic transformation of carbon at the surface), export production (related to the transport of organic material to depth) and the amount of remineralised carbon. On average, ESMs project a decrease in globally averaged primary production and export production across various future scenarios of increasing atmospheric $CO_2$ [Henson et al., 2022, Wilson et al., 2022, Kwiatkowski et al., 2020]. Yet, these results differ regionally with, e.g., a general increase in the Arctic Ocean, Southern Ocean, and a general decrease in the equatorial Pacific [Myksvoll et al., 2023, Henson et al., 2022, Wilson et al., 2022, Kwiatkowski et al., 2020]. Globally and regionally, the range of projected changes in primary production and export production differs largely between ESMs such that the inter-model range of the change is often more than twice its multi-model mean [Tagliabue et al., 2021]. In contrast, more remineralization of organic matter in the interior can be expected due to a more sluggish circulation [Tjiputra et al., 2018, Weijer et al., 2020], increasing the effectiveness of the BCP despite a reduced export production from the surface [Liu et al., 2023]. However, despite model consensus on a global increase of remineralised carbon across scenarios, the amplitude varies widely between models [Wilson et al., 2022].

The quantity of remineralised carbon ($DIC_{remin}$) is a good indicator of the strength of the BCP and its impact on atmospheric $CO_2$ [Marinov et al., 2008, Kwon et al., 2009, Koeve et al., 2020, Frenger et al., 2024]. In a steady state climate, large $DIC_{remin}$ stocks correspond to low atmospheric $CO_2$ levels [Marinov et al., 2008, Frenger et al., 2024] and in a transient climate, the strongest increase in $DIC_{remin}$ corresponds to the strongest biologically-induced decline in atmospheric $CO_2$ [Koeve et al., 2020, Frenger et al., 2024]. In contrast, export production is unrelated to atmospheric $CO_2$ [Marinov et al., 2008, Kwon et al., 2009, Frenger et al., 2024]. $DIC_{remin}$ can be estimated from apparent oxygen utilisation (AOU, Frenger et al. [2024], Wilson et al. [2022]), which measures the deficit of oxygen compared to saturation. It is an estimate of the cumulative oxygen utilised to remineralise organic material since the water parcel was last in contact with the atmosphere. Despite some limitations such as assuming 100% oxygen saturation at the surface [Ito et al., 2004], changes in AOU can be used for quantifying the impact of the BCP on atmospheric $CO_2$ [Koeve et al., 2020].

AOU is traditionally supposed to be the product of the oxygen utilisation rate (OUR) and an estimation of the time since the water-mass was last in contact with atmosphere [Sulpis et al., 2023, Feely et al., 2004, Sarmiento et al., 1990]. In regions with sufficient oxygen concentration, the relation between AOU and age is linear when they are affected similarly by transport [Koeve and Kähler, 2016]. Typically the linear relationship breaks in areas where gradients are too different [Thomas et al., 2020]. A stronger remineralization closer to the surface or below highly productive zones (e.g., equatorial Pacific) will locally increase AOU without any correlation to a change in age. The linear relation between AOU and age has been used to estimate the OUR [Sulpis et al., 2023] and as a proxy of water-mass age [Thomas et al., 2020]. The relationship between trends or changes in



AOU and age can further be used to decompose changes in AOU into circulation-driven and biologically-driven factors. So far
this relationship has been little explored in future climate projection. Bopp et al. [2017] found a strong relationship in one ESM
from the Coupled Model Intercomparison Project Phase 5 (CMIP5), however, ventilation ages were not available for the other
CMIP5 models. More recently, Liu et al. [2023] explored the relation between changes in circulation and changes in AOU.
They found that the slowing down of the meridional overturning circulation, which is an indicator of ocean interior residence
time, would allow more time for the exported biogenic carbon to accumulate at depth and thus increase the deep ocean storage
of carbon by the BCP.
In this work we further explore the relationship between changes in circulation and changes in the BCP using Earth system
model simulations from the sixth Coupled Model Intercomparison Project (CMIP6) as well as observations from the Global
Ocean Data Analysis Project (GLODAPv2, Lauvset et al. [2024]). Following the approach suggested by Frenger et al. [2024],
we use remineralised DIC, measured from AOU, as indicator for the BCP. We show that changes in AOU are linearly related to
changes in age in large parts of the deep ocean. We further use the linear relationship to quantify the respective role circulation
changes in the future evolution of the BCP. Lastly, we discuss future opportunities to constrain the estimates of the deep ocean
BCP with observations.

## Methods

### Earth system models outputs and observational data

Eleven Earth system models (ESMs) provide the monthly 3D output fields required to compute AOU for the preindustrial
control (piControl), historical and SSP5-8.5 future scenario simulations in a replica of the CMIP6 database. Among these
eleven ESMs, only eight also provide outputs for the age tracer: MPI-ESM1.2-LR and MPI-ESM1.2-HR [Mauritsen et al.,
2019], ACCESS-ESM1.5 [Ziehn et al., 2020], IPSL-CM6A-LR [Boucher et al., 2020], MIROC-ES2L [Hajima et al., 2020],
NorESM2-LM and NorESM2-MM [Seland et al., 2020] and CanESM5 [Swart et al., 2019]. We do not consider NorESM2-MM
and MPI-ESM1.2-HR here to keep only one variant of each model. We also do not consider CanESM5 because it does not
provide phosphate fields that are used to compute the PO-tracer [Broecker et al., 1991], required to define water-masses (see
section about water-masses definition). Hence, five ESMs are selected to be analyzed in detail in this work. For comparison,
we also compute AOU for the four remaining ESMs (CanESM5, CNRM-ESM2-1 [Séférian et al., 2019], GFDL-ESM4 [Dunne
et al., 2020], UKESM1-0-LL [Sellar et al., 2020]).
To have an observational baseline over the recent period, we also conduct an observation-based analysis of the trends in AOU
and trends in age using co-located temperature, salinity, phosphate and oxygen measurements as well as AOU and age estimates
from the observational data product GLODAPv2.2024 [Lauvset et al., 2024]. AOU is computed in the same way as for the ESMs
(see next section about AOU and remineralised carbon), while age is derived from measurements of the chlorofluorocarbons
(CFCs) CFC-11 and CFC-12, as well as the transient tracer sulphur hexafluoride (SF6) [Jeansson et al., 2021]. The time range
of the observational baseline is limited by the age dataset and extends from 1972 to 2013. Only observations below 1000 metres
depth are considered to avoid the influence of mixed-layer processes and subtropical gyres.

### Apparent oxygen utilisation and remineralised carbon

Apparent oxygen utilisation (AOU in $[\text{mol} \, O_2 \, \text{m}^{-3}]$) is computed as:

$$\text{AOU} = O_2^{\text{sat}} - O_2 \qquad \qquad \textbf{Equation } 1.$$

where $O_2$ is the in-situ dissolved oxygen concentration and $O_2^{\text{sat}}$ is the dissolved oxygen concentration at saturation computed
from temperature and salinity following Garcia and Gordon [1993, 1992]. The amount of carbon resulting from this remineral-





ization ($\text{DIC}_{\text{remin}}$ in [$\text{g}\,\text{C}\,\text{m}^{-3}$]) is estimated as:
$$\text{DIC}_{\text{remin}} = m_{\text{C}} \times R_{\text{C:O}_2} \times \text{AOU}$$
**Equation** 2.

where $m_{\text{C}}$ is the molecular weight of carbon (12.01 $\text{g}\,\text{mol}^{-1}$) and $R_{\text{C:O}_2}$ is the stoichiometric ratio between carbon and oxygen
(117:170, Anderson and Sarmiento [1994]).
Although providing a reasonably good indication of the BCP strength and its impact on atmospheric $CO_2$ [Koeve et al., 2020,
Frenger et al., 2024], AOU has a couple of pitfalls that should be kept in mind. First, it assumes that at the surface, oxygen
concentration is in equilibrium with the atmosphere. This assumption is valid in most of the ocean, yet in high latitudes, water
parcels can be detrained from the mixed layer while being under-saturated leading to an overestimation of respiration and
AOU, notably in the deep ocean [Ito et al., 2004, Duteil et al., 2013]. True Oxygen utilisation [Ito et al., 2004] or Evaluated
Oxygen utilisation [Duteil et al., 2013] are intended to overcome this limitation. However, the computation of these variables
requires additional tracers (e.g., preformed $O_2$, [Tjiputra et al., 2020]) that are not routinely available in the CMIP6 output
database. Another limitation is that AOU only measures aerobic remineralization. Yet, when oxygen levels are too low,
anaerobic remineralization will take place and use other oxidants (e.g., nitrate for denitrification) instead of oxygen. In the open
ocean, denitrification typically occurs in suboxic waters, when oxygen concentrations drop below 5 $\mu\text{mol}\,O_2\,l^{-1}$ [Keeling et al.,
2010]. Suboxic waters represent only 0.1% of the contemporary ocean and are located in the upper 1000 metres [Deutsch et al.,
2011, Keeling et al., 2010]. During the 21st century, suboxic volume may extend but is projected to not exceed 1% of the ocean
volume [Deutsch et al., 2011, Cocco et al., 2013, Fu et al., 2018].
**Water-masses definition**
We aim to find a linear relationship between the spatial distribution of AOU trends and age trends that is representative for most
of the deep ocean. From now on and unless specified otherwise, we define the deep ocean as the ocean below 1000 metres. We
assess the linear relationship within different water-masses of the deep ocean characterized with a combination of the PO-tracer
($\text{PO}^*$, Broecker et al. [1991]) and density.
For the water-mass definition of the ESMs, neither density nor $\text{PO}^*$ are standard outputs in the CMIP6 database so that we
compute density with the Gibbs SeaWater (GSW) Oceanographic Toolbox of TEOS-10 in xarray [Caneill and Barna, 2024,
McDougall and Barker, 2011] and $\text{PO}^*$ based on the definition by Broecker et al. [1991] ($\text{PO}^* = \text{PO}_4 + O_2/175$). Both variables
are averaged over the years 1972 to 2013, i.e. the time period covered by the observational dataset used in this work. Our water-
mass definition for the ESMs focuses only on the deep ocean below 1000 metres and uses $\text{PO}^*$-thresholds to define water masses
originating in the North Atlantic and Southern Ocean (respectively $0.35 < \text{PO}^* \leq 0.91$ $\text{mmol}\,\text{PO}_4\,\text{m}^{-3}$ and $0.91 < \text{PO}^* \leq 2$
$\text{mmol}\,\text{PO}_4\,\text{m}^{-3}$). These thresholds are derived from Broecker et al. [1998], yet they have been refined by iterative testing to
better suit the ESMs. Only grid-cells located south of 60°N are considered to exclude the Arctic Ocean. For the Atlantic
water-masses, only grid-cells located between 90°E and 30°W are included to exclude grid-cells in the Pacific and Indian ocean
that fulfill the $\text{PO}^*$-threshold. Each of these water-masses is then split into half according to density (see supplementary Tab.
S1), leading to four water-masses: (i) the Atlantic light waters, (ii) the Atlantic dense waters, (iii) the Southern light waters, and
(iv) the Southern dense waters. These four water-masses cover between 90 % and 98 % of the entire deep ocean, depending on
the ESM. For each water-mass and each ESM we define a spatial mask (supplementary Fig. S1), which is used to identify grid
points belonging to the same water-mass and compute the linear regression between trends in AOU and trends in age (see next
section). We keep the masks constant throughout the historical and SSP5-8.5 simulations as the masks show minimal sensitivity
to the time period used for creating the $\text{PO}^*$ and density fields (supplementary Figs. S1 and S2)
Similar to the definition of water-masses used for the ESMs, observations are classified into water masses originating in the
Southern Ocean and North Atlantic based on their $\text{PO}^*$ values (supplementary Fig. S3). Waters originating in the South-
ern Ocean are defined via $1.2 \leq \text{PO}^* \leq 2.0$ $\mu\text{mol}\,\text{PO}_4\,\text{kg}^{-1}$ and those originating in the North Atlantic Ocean via $\text{PO}^* < 1.2$
$\mu\text{mol}\,\text{PO}_4\,\text{kg}^{-1}$ with $\text{PO}^*$ thresholds based on Broecker et al. [1998]. In lighter density classes, due to the fanning out of tem-





perature and salinity, the number of data is too low to identify a relationship between AOU trends and age trends. Thus, the
resulting water-masses are already relatively dense, with an average of $\sigma_0 = 27.7\,\mathrm{kg\,m^{-3}}$ and density no lighter than $27\,\mathrm{kg\,m^{-3}}$
and were not further separated into light and dense waters. The water-masses will be referred to as Southern dense waters and
North Atlantic dense waters to facilitate a meaningful comparison with their model counterparts and are most representative of
the water-mass end members.

**Relationship between trends in AOU and trends in age**

Just as the relationship between AOU and age can be linear [Sulpis et al., 2023], one might expect that the trends in AOU and
the trends in age can be linearly related. In this work we intend to express the trends in AOU ($\frac{\mathrm{dAOU}}{\mathrm{dt}}$) via trends in age ($\frac{\mathrm{dage}}{\mathrm{dt}}$) as
follows:

$$\frac{\mathrm{dAOU}}{\mathrm{dt}} = S_{\Delta\mathrm{age}}^{\Delta\mathrm{AOU}} \times \frac{\mathrm{dage}}{\mathrm{dt}} + B + \varepsilon. \qquad \textbf{Equation } 3.$$

We assess the linear relationship between spatial fields of AOU trends and age trends within the previously define water-masses
using a linear regression [Virtanen et al., 2020]. The slope of the linear regression is the sensitivity of AOU changes to age
changes ($S_{\Delta\mathrm{age}}^{\Delta\mathrm{AOU}}$), the intercept is $B$ and $\varepsilon$ is the error of the linear regression. $B + \varepsilon$ gathers changes in AOU that are not linearly
related to changes in age such as changes in remineralization rates. $S_{\Delta\mathrm{age}}^{\Delta\mathrm{AOU}}$ defined here is connected to the oxygen utilisation
rate (OUR) defined in other studies [Sulpis et al., 2023, Feely et al., 2004]. Indeed, if the equation AOU = OUR × age is
differentiated with respect to time, then $S_{\Delta\mathrm{age}}^{\Delta\mathrm{AOU}}$ and OUR are a similar quantity: an estimate of a spatio-temporal average of the
local instantaneous oxygen utilisation rate. We choose to call the slope of the linear regression $S_{\Delta\mathrm{age}}^{\Delta\mathrm{AOU}}$ instead of OUR for two
reasons: 1) we think this word convey more accurately the purpose of the analysis, i.e. investigating the relationship between
AOU trends and age trends and 2) we want to avoid ambiguity with studies working at estimating OUR (e.g. Sulpis et al.
[2023]).
For the analysis of the ESMs, it is crucial to estimate and remove the drifts in the simulated fields of AOU and age tracer
before calculating their respective trends. The trends and drifts are estimated for every ocean grid-cell of the ESMs using a
linear regression over years 1850 to 2099 of the piControl simulation (note that actual years are adjusted to fit 1850-2099, see
supplementary Tab. S2). Outputs from the historical and SSP5-8.5 simulations are then drift corrected for each point in time
$t$ ($X_{\mathrm{drift\text{-}corrected}}(t) = X_{\mathrm{drift\text{-}uncorrected}}(t) - (t - t_0) \times drift$, with $t_0$ referring to 1850) before computing the trends. The trends are
computed for the years (i) 1972-2013 of the historical simulation to match the time period of available observational data and
(ii) 2015-2099 (the entire SSP5-8.5 simulation). When considering the ESMs, for the time period 1972-2013, between 49 %
(NorESM2-LM) and 72 % (IPSL-CM6A-LR, MIROC-ES2L) of the deep ocean grid points have significant trends (p-value ≤
0.05) in both AOU and age, while between 84 % (NorESM2-LM) and 94 % (MIROC-ES2L) have significant trends for the
time-period 2015-2099 (see supplementary Fig. S4). The non-significant trends are very close to zero. For each ESM, we only
consider grid-points with significant trends for computing the linear regression in each water-mass.
To overcome the difficulty of identifying trends with highly spatio-temporally sparse observational data, as it is the case for age
estimates, we collapse the available observations in Temperature-Salinity (T-S) space. Trends in AOU and age are computed
within bins in the T-S space (supplementary Fig. S5). Temperature ranges between −1.5 and 8 °C and is divided into 350
bins while salinity ranges between 34.2 and 34.9 and is divided into 300 bins. The chosen resolution of the T-S-bins, about
0.027 °C and 0.0023, comes from the trade-off between: (i) ensuring that comparable measurements are grouped into the same
T-S-bin even if they have a different geographical location, (ii) providing enough data points per T-S-bins to find significant
(p-value ≤ 0.05) trends and (iii) providing enough trends estimates (one per T-S-bins) to identify a significant (p-value ≤ 0.05)
relationship between AOU trends and age trends. Trends for AOU and age, (dAOU/dt) and (dage/dt) are computed when five
or more observations are grouped into a given T-S bin. Trends are then grouped into the Southern and Atlantic water-masses
defined previously. For the Southern water-mass, out of 1076 trends identified, 77 are significant for age and 107 for AOU.
For the Atlantic water-mass, 1816 trends are identified, out of which 188 are significant for age and 246 for AOU. When
accounting only for the co-located measurements that have significant trends both in age and AOU, the number of trends is



reduced to 23 and 118 for the Southern and Atlantic water-masses, respectively. As in the modeling counterpart of the analysis,
a linear regression was computed between the spatial distributions of AOU trends and the age trends, which is then used as an
observational constraint.
In this work we apply linear regressions for estimating trends and evaluating linear relationships. In these applications, we
evaluate the significance of the trends and the linear relationship base on the p-values testing the null hypothesis of zero slopes,
i.e. no trends or no linear relationship. When the p-value is lower than or equal to 0.05 we consider the trends or the linear
relationship to be significant. The linear regression also provides a 95 % confidence interval for the slope, serving as a measure
of the uncertainty associated with $S_{\Delta\text{age}}^{\Delta\text{AOU}}$. If this uncertainty is not specifically stated, it means that it is negligible with respect
to the number of significant figures provided.

## Results

Contemporary spatial AOU patterns reflect physical transport and biological oxygen consumption. For example, AOU is par-
ticularly high in areas combining weak ventilation or ventilation of oxygen-depleted water-masses and intense remineralization
such as the deep ocean, the North Pacific or in the upper 1000 metres in the equatorial band (Fig. 1a,b,c). Earth system models
(ESMs) reproduce the general patterns shown by observations (supplementary Fig. S6), yet with some regional strong positive
and negative biases relative to observations (Fig. 1d,e,f). On average, ESMs overestimate AOU in the ocean deeper than 2000
metres north of ca. 40°S and below 1000 metres in the Pacific north of ca. 50°S. In contrast, ESMs underestimate AOU in the
Southern Ocean (south of ca. 30°S) and above 1000 metres depth in the whole northern hemisphere. In addition to these biases
in the model-mean, we note that there is a strong inter-model spread in large parts of the ocean, where the range of ESM values
is higher than 70 % of the observation value (stippling in Fig. 1d,e,f and supplementary Fig. S7).





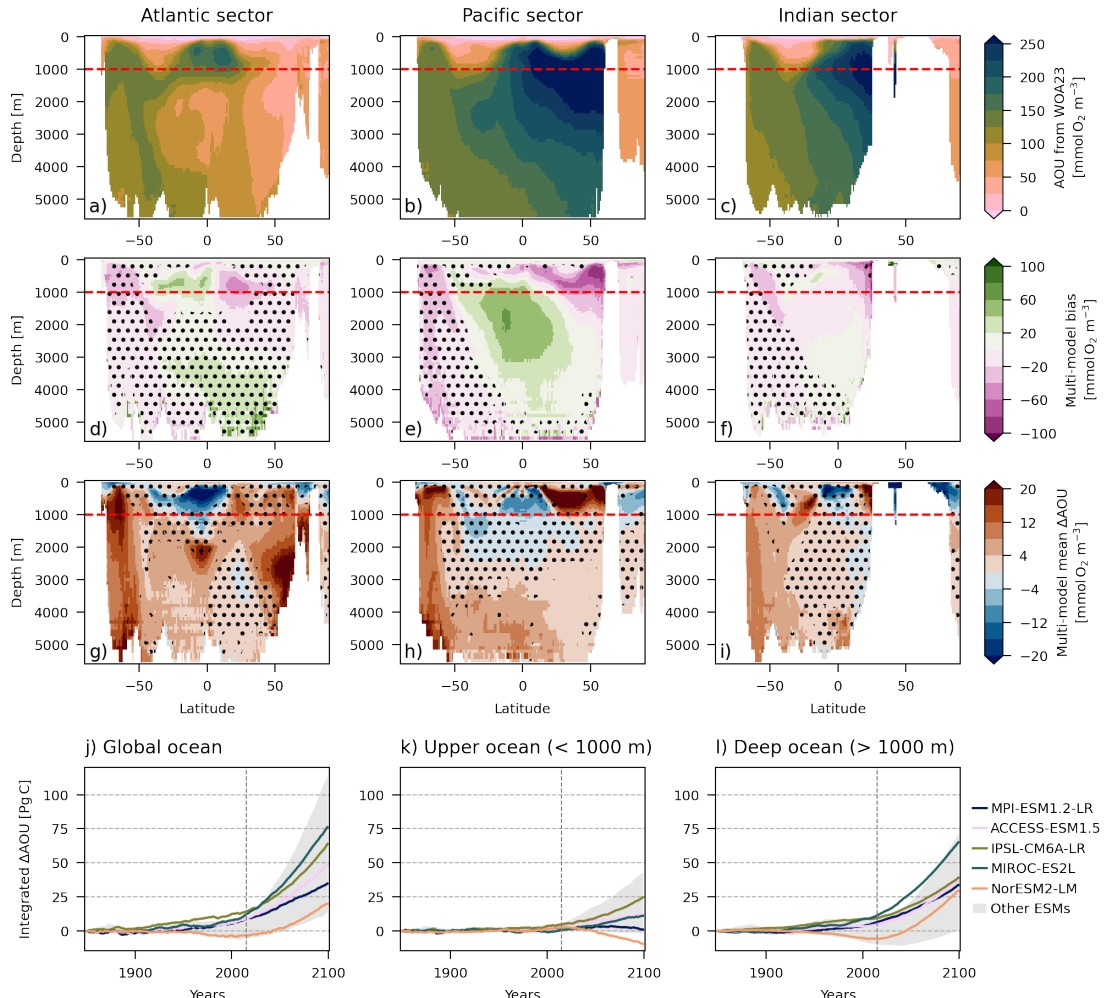

**Figure 1.** Evaluation of apparent oxygen utilisation (AOU) and consistency in the projected future change (ΔAOU) as simulated by Earth system models (ESMs). (a, b, c) AOU from the World Ocean Atlas 2023 (WOA23, Garcia et al. [2024]), averaged over 1971-2000 in the Atlantic (10°W to 60°W), Pacific (130°W to 180°W) and Indian (40°E to 90°E) sectors of the ocean. (d, e, f) Multi-model mean of AOU bias against WOA23. Stippling shows AOU uncertainty in ESMs, i.e. when the range between the highest and lowest ESM values is greater than 70 % of the WOA23 value. Refer to supplementary Fig. S7 for individual ESM bias. (g, h, i) Multi-model mean of projected change (1971-2000 minus 2070-2099) under the SSP5-8.5 scenario, zonally average. Stippling shows ΔAOU uncertainty in ESMs, i.e. when the range between the strongest and weakest ΔAOU exceed three times the multi-model mean ΔAOU. Refer to supplementary Fig. S8 for individual ESM ΔAOU. The red dashed lines indicate the 1000 metres depth separating the upper and deep ocean. (j, k, l) Time series of ΔAOU integrated on the (j) global, (k) upper and (l) deep ocean for each ESM considered in this study. The gray shading shows the range of other ESMs not used in this study (CanESM5, CNRM-ESM2-1, GFDL-ESM4, UKESM1-0-LL). The vertical dashed gray line (year 2015) separates the historical and SSP5-8.5 scenarios.

In the majority of the ocean, AOU is projected to increase under the SSP5-8.5 scenario (Fig. 1g,h,i). Most of the increase
occurs below 1000 metres (Fig. 1l), with agreement on the sign of change among the models (supplementary Fig. S8). Above
1000 metres, AOU is projected to decrease in areas around the Equator or near the surface in the high latitude (Fig. 1g,h,i).
The uncertainty of the projected change is considerable between ESMs with the inter-model spread exceeding three times the
multi-model mean in the intermediate depth of the Pacific, in the low-latitude Indian, and in the deep subtropical North Atlantic
(stippling in Fig. 1g,h,i and supplementary Fig. S8). By 2099, the integrated projected global change in AOU compared to 1850



ranges from 20 to 76 PgC (Fig. 1j). A substantial share of the inter-model uncertainty in AOU changes stems from the deep ocean (below 1000 metres). Here, the ESM spread encompasses 20 to 65 PgC (Fig. 1l), while it ranges from $-10$ to 25 PgC above 1000 metres (Fig. 1k). The inter-model differences in AOU changes within the ESMs used in this study is representative of the inter-model differences in the AOU changes as seen by other ESMs (grey shading in Fig. 1j,k,i).

For our model ensemble and the grid-points with significant trends in AOU and age in our defined Atlantic and Southern water-masses, the AOU trends are significantly correlated with the age trends for the years 1972-2013 (Fig. 2) and 2015-2099 (supplementary Fig. S9). The coefficient of determination, $R^2$, is higher than 0.7, indicating that the spatial variability in age trends can explain more that 70 % of the spatial variability in AOU trends, for four out of five ESMs in the dense water-masses and three out of five ESMs in the light water-masses. We only find weaker correlations between AOU trends and age trends in five cases (see $R^2 < 0.7$ in Fig. 2). This is probably related to the definition of the different water-masses not being fully appropriate for all models. For example, some deep grid points counted towards the Atlantic light waters could arguably be included in the Atlantic dense waters. In addition, some of the grid points defined as light waters are situated around 1000 metres and could be part of sub-surface waters such as the Antarctic Intermediate Waters or Subantarctic Mode Water. The linear regression analysis cover only between 49 % and 72 % of the deep ocean for the 1972-2013 period, depending on the ESM, because i) we only consider grid points with significant trends in age and AOU, and ii) large part of the deep ocean have weak and non-significant trends during the contemporary period. For the 2015-2099 period, the linear regression analysis cover between 79 % and 95 % of the deep ocean, depending on the ESM, as the trends are stronger and more significant (supplementary Fig. S9 and Fig. S4). The inclusion of the non-significant trends decreases $R^2$ but does not substantially alter the slope of the linear regression (supplementary Fig. S10 and Fig. S11). A positive linear correlation is also found between the significant trends in AOU and age from the observation dataset (supplementary Fig. S12). Here, $R^2$ is 0.88 and 0.92 for the Southern and Atlantic water-masses, respectively, suggesting that about 90 % of the spatial variability in AOU trends can be explained by the spatial variability in age trends for the few significant trends in the Southern and Atlantic water-masses.




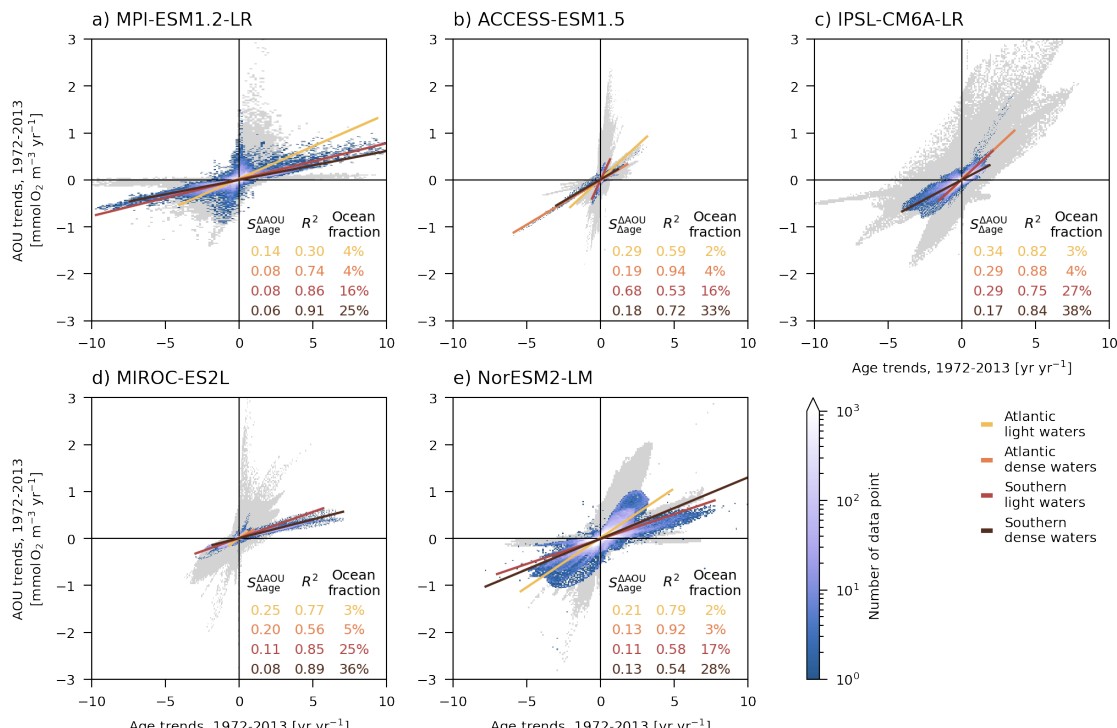

**Figure 2.** Distribution of the trends in age and trends in apparent oxygen utilisation (AOU) for the contemporary period (1972-2013) simulated with five Earth system models (ESMs): a) MPI-ESM1.2-LR, b) ACCESS-ESM1.5, c) IPSL-CM6A-LR, d) MIROC-ES2L, e) NorESM2-LM. The blue shading shows the number of data point for each bin of age trends and AOU trends for the Southern and Atlantic light/dense waters. A linear regression is computed between the AOU trends and age trends for each water-mass. On each panel, the slope ($S_{\Delta age}^{\Delta AOU}$), the coefficient of determination ($R^2$) and the fraction of the deep ocean volume are shown in different colours for each water-masse. The gray shading show the distribution of trends for the entire ocean.

The simulated sensitivities of AOU change to age change ($S_{\Delta age}^{\Delta AOU}$) are relatively similar in the light and dense waters. Yet, as might be expected, given that remineralization is stronger in the shallower regions, $S_{\Delta age}^{\Delta AOU}$ is slightly stronger in light waters. We find that ESMs with a large (small) $S_{\Delta age}^{\Delta AOU}$ in the contemporary period (1972-2013) also have a large (small) $S_{\Delta age}^{\Delta AOU}$ for the future period under the SSP5-8.5 climate change scenario (Fig. 3). In addition, for each ESM, the future $S_{\Delta age}^{\Delta AOU}$ is similar or smaller than the contemporary one (except for three cases, points above the 1:1 line in Fig. Fig. 3), suggesting a reduction of the remineralization rate under the SSP5-8.5 climate change scenario. The linear relation between present and future $S_{\Delta age}^{\Delta AOU}$ is strong for the Southern light and dense waters and the Atlantic dense waters across our model ensemble, as indicated by a linear regression giving coefficients of determination higher than 0.7 and p-values below 0.05 (Fig. 3). The only exception are the Atlantic light waters with a weak linear relation across the model ensemble, mainly due to the behavior of NorESM2-LM. $S_{\Delta age}^{\Delta AOU}$ evaluated from the observations are $0.11 \pm 0.02$ and $0.04 \pm 0.01$ mmol $O_2$ m$^{-3}$ yr$^{-1}$ for the Atlantic dense and Southern dense water-masses, respectively (vertical blue line and shading in Fig. 3). They have not been split into light and dense waters due to the limited number of data points (see methods). In both cases, the values are on the low side of the ESMs range: 0.08 to 0.29 mmol $O_2$ m$^{-3}$ yr$^{-1}$ and 0.06 to 0.18 mmol $O_2$ m$^{-3}$ yr$^{-1}$ for the Atlantic dense and Southern dense water-masses, respectively.





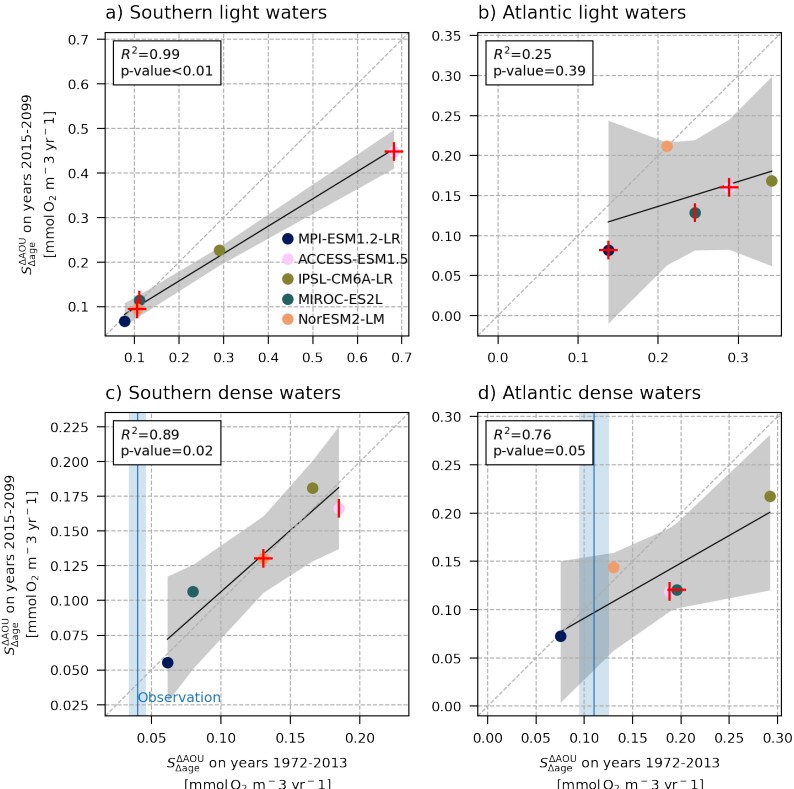

**Figure 3.** Distribution of the sensitivity of AOU change to age change ($S_{\Delta age}^{\Delta AOU}$) in each water-masse: a) Southern light, b) Atlantic light, c) Southern dense, d) Atlantic dense. Each dot shows the $S_{\Delta age}^{\Delta AOU}$ for one Earth system model (ESM) on the contemporary (1972-2013) and future (2015-2099) period. For few models, the red horizontal/vertical dash indicates a weak correlation ($R^2 < 0.7$) between AOU trends and age trends for the contemporary/future period. The black line shows the linear regression and the gray shading its confidence interval. The associated coefficient of determination ($R^2$) and the p-value are indicated in each panel. The diagonal dashed gray line is the 1:1 line. The vertical blue lines (and shading) show the $S_{\Delta age}^{\Delta AOU}$ (and its uncertainty) on years 1972-2013 estimated from observations of the GLODAPv2 database, only for the Southern and Atlantic dense water-masses.

Once a linear relationship has been established providing the average trend in AOU for a given trend in age, we can use it to further quantify the contribution of age trends to the trends in AOU in each ESM. This contribution is estimated by multiplying the age trends by $S_{\Delta age}^{\Delta AOU}$ estimated in each of the four water-masses ($S_{\Delta age}^{\Delta AOU} \times \frac{dage}{dt}$ in Eq. 3). Globally integrated, age trends contribute from 38 % (ACCESM-ESM1.5) to 95 % (IPSL-CM6A-LR) of the AOU trends in these water-masses (Fig. 4a). In general, the positive trends in AOU mostly arise from the Southern dense water-mass, and are driven by positive trends in age (Fig. 4b-p). The Atlantic dense water-mass exhibits also intense local positive AOU trends driven by age trends. In these two ventilation regions, the models suggest a weakening in the ventilation rates in the future (increasing age). In contrast, negative AOU trends are mostly located in the Southern and Atlantic light water-masses, found between 1000 and 2000 metres depth in the subtropics and equatorial region. In these areas, negative age trends play a major role indicating that waters get younger because of a shift in water-mass structure or stronger ventilation, thought stronger ventilation seems less likely considering that stratification increase everywhere in the ocean [Kwiatkowski et al., 2020]. The remainder term, $B + \varepsilon$, locally either slightly compensates or reinforces changes driven by age trends resulting globally in a positive contribution to AOU trends. MPI-ESM1.2-LR, the ESM with the lowest $S_{\Delta age}^{\Delta AOU}$ and closest to the observation estimate, has a water-mass averaged coefficient of determination, $\overline{R^2}$, of 0.78 and estimate the contribution of age trends to 57 % (Fig. 4a). MIROC-ES2L and NorESM2-LM, whose $S_{\Delta age}^{\Delta AOU}$ is also relatively close to the observation estimate for at least one of the water-masses, have higher contribution of age trends. However, $\overline{R^2}$ is below 0.7 for these two models. MPI-ESM1.2-LR and MIROC-ES2L are quite contrasted in



terms of global trends in AOU and contribution from age trends. In addition, MPI-ESM1.2-LR simulates strong AOU trends, driven by age trends, in the Atlantic dense waters north of 50°N, between 2000 and 3000 metres depth, a feature not present in MIROC-ES2L (Fig. 4b,ck,l). The significant trends in the four water-masses cover at least 79 % of the deep ocean. Integrated over these water-masses, the AOU trends are very close to AOU trends integrated on the entire deep ocean (dark grey bars versus black contour bars in Fig. 4a) and represent a significant part of the globally integrated AOU trends (dark grey bars versus light grey bars in Fig. 4a).



# Discussion and conclusions

Understanding changes in ocean BCP and its impact on future climate change remains an outstanding research question [Tjiputra et al., 2025]. In this work, we have demonstrated that the spatial fields of AOU trends (an indicator of BCP) and age trends are strongly correlated in the ocean deeper than 1000 metres where spatial variability in age trends can explain at least 70 % of the spatial variability in AOU trends ($R^2 \geq 0.7$). This relationship is identified in simulations of the contemporary period (1972-2013) and simulations of the future period (2015-2099) under the SSP5-8.5 climate change scenario. The sensitivity of AOU change to age change, $S_{\Delta\text{age}}^{\Delta\text{AOU}}$ (that is, the slope of the linear regression), varies between the ESMs and the water masses from 0.06 to 0.34 $\text{mmol}\,O_2\,\text{m}^{-3}\,\text{yr}^{-1}$ for the contemporary period, when considering only the linear relationship associated with $R^2 \geq 0.7$. $S_{\Delta\text{age}}^{\Delta\text{AOU}}$ remain relatively similar when computed for the 2015-2099 period. Using the linear relationship we estimate that, on the 2015-2099 time period, the increase in age due to changes in circulation or ventilation rates contribute between 38 % and 95 % to the increase in deep ocean $\text{DIC}_{\text{remin}}$ depending on the ESM.

Our estimates of $S_{\Delta\text{age}}^{\Delta\text{AOU}}$ derived from observational data for the contemporary period are consistent with the lower range of estimates produced by ESMs. The reliability of these observational estimates for evaluating ESMs is compromised by the limited number of data points used in the estimation. However, increasing the number of data points by including non-significant trends yields only to minor changes in the estimations. In addition, given that $S_{\Delta\text{age}}^{\Delta\text{AOU}}$ is similar to some extent to an estimation of the oxygen utilisation rate (OUR) averaged within the water-masses considered, comparison with prior OUR estimates is appropriate. In the deep ocean, OUR estimations typically vary around 0.1 $\text{mmol}\,O_2\,\text{m}^{-3}\,\text{yr}^{-1}$ [Sulpis et al., 2023]. These observation-based estimations are also consistent with the lower range of the ESMs estimations. This suggests that the changes in simulated interior AOU may be overly sensitive to changes in age in most of the models analysed here. In addition, our analysis shows a relative stability over time of this sensitivity. This is likely an underestimation of the sensitivity on the future warmer period since observational studies has suggested increase remineralization and oxygen consumption with higher temperature [Brewer and Peltzer, 2017]. These results indicates that the projected increase in AOU and thus $\text{DIC}_{\text{remin}}$ might be weaker for a similar increase in age.

Our results highlight the importance of circulation changes on the changes in AOU and therefore on $\text{DIC}_{\text{remin}}$ in the deep ocean. Previous studies suggested that circulation was the main driver of changes in interior carbon content during the past and future climate [Bopp et al., 2017, Kessler et al., 2018, Liu and Primeau, 2023]. We quantify that between 2015 and 2099, under the SSP5-8.5 climate change scenario, in the ocean below 1000 metres, an increase in age contributes between 57 % and 81 % to an increase $\text{DIC}_{\text{remin}}$, based on MPI-ESM1.2-LR, MIROC-ES2L and NorESM2-LM whose $S_{\Delta\text{age}}^{\Delta\text{AOU}}$ are the closest to the observation-based estimates. The densest water-mass coming from the Southern ocean (southern dense water-mass) contribute predominantly to the deep ocean $\text{DIC}_{\text{remin}}$ increase. This water-mass covers a large portion of the deep ocean, have particularly strong correlation between spatial fields of AOU trends and age trends and the contribution from the age increase is even larger. While we highlight the importance of change in age in this water-mass, a substantial portion of the change in AOU is not driven by change in age in lighter water-masses, and question the reliability of using change in age as a proxy for change in AOU in the ocean above 2000 metres. Here, changes in export [Henson et al., 2022], spatially variable oxygen utilisation rate [Sulpis et al., 2023] can de-correlate changes in AOU from changes in age.

One caveat of our work is the use of AOU as a proxy of remineralised organic matter, notably as we focus on the deep ocean where water parcels coming from the high latitude can be exported while being-undersaturated with respect to oxygen [Ito et al., 2004, Duteil et al., 2013]. Interestingly, when compared to true oxygen utilisation (TOU) that is a more accurate measure of remineralised organic matter, AOU overestimates TOU but changes in AOU underestimates changes in TOU by 25 % [Koeve et al., 2020]. For our work, this suggests 25 % stronger trends in remineralised organic matter and thus stronger sensitivity of the BCP to circulation slow down relative to the sensitivity estimated from AOU trends in our ESM ensemble and the observation baseline. In addition, AOU underestimates organic matter remineralisation because it does not account for denitrification occurring in suboxic waters. In global warming simulations, the volume of suboxic waters increases all along the 20th and 21st century resulting in a small increase in denitrification [Fu et al., 2018, Cocco et al., 2013]. Nevertheless, since



suboxic waters are mostly located in the upper 1000 metres of the ocean, the omission of denitrification is expected to have a
minimal impact on our results. If it does have an impact, it would likely result in a small underestimation of $S^{\Delta\text{AOU}}_{\Delta\text{age}}$.
One of the initial motivation for this work was to constrain ESMs projections of AOU based on the sensitivity of AOU changes
to age changes. The linear relationship between present and future sensitivity across ESMs is promising. It can, in theory, be
used to identify ESMs whose sensitivities are the most consistent with observations in the contemporary period and be used
to constrain the sensitivity of the future period. However, at this point in time, we cannot directly constrain AOU projections
following an emergent constraint approach [Bourgeois et al., 2022, Kwiatkowski et al., 2017] because of the small ESM en-
semble available and the limitation of the currently available observations. Nonetheless, below 2000 metres, our results suggest
that if we can constrain deep ocean ventilation changes then we can constrain projections of deep ocean AOU. However, to
identify the best ESMs at projecting deep ocean ventilation changes is challenging. For instance, under a different climate,
the last glacial maximum, ESMs simulate very different changes in Atlantic MOC (meridional overturning circulation) depth
and strength one from another and none of them is really consistent with the estimations from proxies [Sherriff-Tadano and
Klockmann, 2021]. On the other hand, simulated changes in the North Atlantic circulation during stadial-interstadial climate
transition show promising comparison with proxies data [Waelbroeck et al., 2023] In addition, constraining only the changes
in the MOC may not be enough to identify best ESMs at projecting changes in AOU in the interior ocean. The uncertainties
associated to interior ocean remineralization in the different models remain. For instance, the slow down of the Southern and
Atlantic MOC in MPI-ESM1.2-LR is stronger compared to MIROC-ES2L [Liu et al., 2023], yet MPI-ESM1.2-LR shows the
weakest change in age-driven AOU trends and MIROC-ES2L the strongest one (Fig. 4). An accurate projection of the carbon
sequestration by the BCP in the deep ocean needs an accurate formation of the deep water masses in the North Atlantic and
Southern Ocean, yet it is not possible to determine even one CMIP6 model that represents those accurately [Heuzé, 2021].
It is our hope that the ESMs represented in CMIP7 will offer further improvements compared to CMIP6 in terms of their
representation of ventilation, especially deep water formation. We would like to emphasize that there is a need for more CMIP7
ESMs to run the simulations with an age and preformed tracers and to make the related outputs available in the CMIP7 database.
Given a larger model ensemble and more observations with ongoing time, our approach is a promising solution that would allow
us to constrain the remineralised carbon sequestration in the deep ocean for the next CMIP7 generation to come.

# Acknowledgements

This work was funded by the European Union under grant agreement no. 101083922 (OceanICU). Views and opinions ex-
pressed are however those of the author(s) only and do not necessarily reflect those of the European Union or European
Research Executive Agency. Neither the European Union nor the granting authority can be held responsible for them. The
computational and storage resources were provided by Sigma2 - the National Infrastructure for High Performance Computing
and Data Storage in Norway (project no. NN1002K, NS1002K). The authors acknowledge the World Climate Research Pro-
gramme, which, through its Working Group on Coupled Modelling, coordinated and promoted CMIP6. The authors thank the
climate modelling groups for producing and making available their model output, the Earth System Grid Federation (ESGF)
for archiving the data and providing access, and the multiple funding agencies who support CMIP6 and ESGF. The authors ac-
knowledge the KeyCLIM project (grant 295046 from the Research Council of Norway) for coordinating access to the CMIP6
data.

# Data availability

CMIP6 outputs are available from the Earth System Grid Federation (ESGF) portals (e.g. https://esgf-node.ipsl.upmc.fr).



## Code availability

The code for producing the figure is available at *https://github.com/damiencouespel/scripts-article-biological-carbon-pump-aou-trends-vs-age-trends*.

## Contributions

Funding acquisition JT, Conceptualization and methodology DC, NG, SKL, JT, Formal analysis and visualization DC, XD, Analysis of the results DC, XD, NG, EJ, SKL, JT, Writing (original draft preparation) DC, Writing (review and editing) DC, XD, NG, EJ, SKL, JT,

## Competing interests

The authors declare no competing interests.

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





**Figure 4.** Trend in apparent oxygen utilisation ($\frac{\mathrm{dAOU}}{\mathrm{dt}}$) and the contribution from trends in age ($S^{\Delta\mathrm{AOU}}_{\Delta\mathrm{age}} \times \frac{\mathrm{dage}}{\mathrm{dt}}$) under the SSP5-8.5 climate change scenario simulated with five Earth system models (ESMs): MPI-ESM1.2-LR, ACCESS-ESM1.5, IPSL-CM6A-LR, MIROC-ES2L, NorESM2-LM. The remainder ($B + \varepsilon$) is computed as the difference between the two aforementioned components (see Eq. 3). Panel a) shows the trends spatially integrated over the global ocean (light grey), the deep ocean (black contour) and the four water-masses considered in this study (dark grey, dark blue and light pink). The white percentage in the dark blue bars indicates the share of the age trends contribution to AOU trends integrated on the four water-masses. At the bottom of the panel, 'Vol. ratio' indicates the share of the deep ocean volume covered by the four water masses, while $\overline{R^2}$ indicates the volume-weighted average of the coefficient of determination over the four water masses. Panels b-p) shows the zonally average trends for each ESM (in rows): first column displays $\frac{\mathrm{dAOU}}{\mathrm{dt}}$, the second column $S^{\Delta\mathrm{AOU}}_{\Delta\mathrm{age}} \times \frac{\mathrm{dage}}{\mathrm{dt}}$ and the third one the remainder $B + \varepsilon$. Trends are computed for the period from 2015 to 2099.





**Figure 4.** (continued)

