# Peer review of "Earth system models overestimate the sensitivity of apparent oxygen"

_EGUsphere, 2025_

## Referee Comment (RC2)

**REVIEW PAPER : EARTH SYSTEM MODELS OVERESTIMATE THE SENSITIVITY OF APPARENT OXYGEN UTILISATION TO AGE CHANGE IN THE DEEP OCEAN**

This study investigates the relationship between apparent oxygen utilisation (AOU) trends and water mass age trends in the deep ocean using Earth system models and observational data from GLODAPv2. The authors establish linear relationships between these variables to assess the biological carbon pump's sensitivity to circulation changes under contemporary and future climate scenarios. While this work addresses an important aspect of ocean biogeochemistry relevant to climate projections and would be of interest to the journal, several aspects require further clarification or investigation to enhance the paper's overall strength.

**Major Comments**

- **Water Mass Definition Methodology**: The water mass classification using PO* thresholds appears somewhat arbitrary and model-dependent. The authors mention "iterative testing" to refine thresholds but provide insufficient detail about this process. Since most of the results are based on this classification, I belive a more detailed section about the derivation of these thresholds is necessary. I also dont understand why there is a difference between the number of water mass definitions for ESMs and observations (four vs two water masses).

- **Linear Relationship Assumption Validity:** While the authors acknowledge in the introduction that linear relationships between AOU and age can break down in certain regions and under specific conditions of the BCP. It would be interesting to further discuss under which conditions this linearity assumption fails in a transient climate scenario. For example, it is unclear to me under which conditions this linear relation still applies (regionally or globally) on SSP5-8.5.

- **Drift Correction Methodology:** The drift correction procedure using a pi-Control simulation makes sens, however, it is not specified how the actual years of the pi-control are adjusted to fit 1850-2099.

- **Observational Data Limitations and T-S Binning**: The T-S space binning approach for observational data analysis is not well justified. The choice of bin resolution (0.027 and 0.0023) seems arbitrary, and the methodology results in very small sample sizes (only 23 and 118 significant joint trends for Southern and Atlantic water masses). The authods should better emphasise the reason behind using this TS binning.

- **On the significance of the observed ovserestimation in ESMs:** The authods show clearly that observational values are "on the low side of the ESMs range" which indicates overestimation of the sensitivity of AOU to age change. While they briefly acknowledge limitations in computing estimates for the contemporary period based on observations (lines around 279), I'm not totaly convinced that the overestimation in ESMs is not due to having more data. For instance, if the ESMs were sampled at the observational locations and the analysis repeated with the same methodological choices as for observations data (e.g. same watter mass definition, same trend calculation approach with the same TS bining), are we going to still going to observe an ovserestimation ?

I hope that the authors will understand my comments in a constructive way, and that I value their work and the time they invested in the preparation of the manuscript. It might be that I have misunderstood something, in this case, if something wasn't clear for me as a reviewer, it is possible that it wouldn't be clear also for the readers.

---

## Author Comment (AC2)

**Authors' Response to Reviews of**

**Earth system models overestimate the sensitivity of apparent oxygen utilisation to age change in the deep ocean**

\*Damien Couespela, Xabier Davilaa, Nadine Gorisa, Emil Jeanssona, Siv K. Lauvseta, Jerry Tjiputraa

aNORCE Research AS, Bjerknes Centre for Climate Research, Bergen, Norway

\*Corresponding author: daco@norceresearch.no

Biogeosciences, EGUsphere [preprint], https://doi.org/10.5194/egusphere-2025-2566

**RC:** Reviewers' Comment, AR: Authors' Response, ☐ Manuscript Text

We thank the two anonymous referees for their thoughtful comments and suggestions. Below are our responses to each comment, detailing the new analysis and modifications that will be made to the manuscript. In particular, we will conduct additional analyses to address uncertainties in the observation-based estimates of the sensitivity of AOU change to age change. The revised manuscript will report on these new analyses.

**1. Reviewer #1**

RC: Summary: Couespel et al. investigated the relationship between apparent oxygen utilization (AOU) and water mass age using both observational data and five Earth system models (ESMs). They found that ESMs tend to overestimate the sensitivity of deep-ocean AOU to changes in ocean ventilation (water mass age) when compared to observations. This overestimation suggests that the models may be overpredicting the future strengthening of the Biological Carbon Pump (BCP) based on AOU trends. Overall, the manuscript is well-written and presented in a clear, accessible manner. The research question addressed is of significant importance to the scientific community. I appreciate the considerable effort the authors have invested. However, I have major concerns regarding the manuscript in its current form that need to be addressed. I recommend a major revision.

**1.1. Major comments**

RC: I have two major concerns regarding the methodology employed in this study. First, the authors calculate  $S_{\Delta age}^{\Delta AOU}$  using ideal age and AOU outputs from the models, while for observational data, they use the mean age of IG-TTD and AOU. I assume that the calculations for observations are based on single-tracer constrained IG-TTD with assumptions such as  $\Delta/\Gamma=1$  and 100% saturation? Please clarify this in the Methods section. Nevertheless, a recent study from Guo et al. (2025) suggests that the mean age of IG-TTD derived from single-tracer can be biased towards younger ages due to imperfect assumptions about the shape of the transit time distribution and the short atmospheric history of abiotic transient tracers like CFC-12 and SF6. Additionally, the trend in the mean age of these tracers can be influenced by uncertainties in their mixing ratios, potentially producing spurious trends. I recommend calculating  $S_{\Delta age}^{\Delta AOU}$  in the models using both AOU and the simulated CFC-12 (or SF6) to ensure methodological consistency. This approach would also allow testing the robustness of the results. To my knowledge, the model you used includes CFC-12 and SF6 simulations, which could be leveraged for this purpose.

AR: As assumed by the reviewer, for the observational data, the mean age from the TTD calculation is estimated using a single tracer, assuming full (100%) saturation when subducted as well as a balance between advection

and mixing,  $\Delta/\Gamma=1$ . We will add these details in the methods sections of the manuscript.

In the ESGF database, CFC-12 and SF6 data are only available for IPSL-CM6A-LR and NorESM2-LM. Thus, age trends estimation using the TTD approach cannot be applied to all models. In addition, as noted by the reviewer and demonstrated by multiple studies, the TTD approach introduces uncertainties in water-mass age estimation. Therefore, we consider that it is more reliable to use the ideal age provided in the models. Nonetheless, we acknowledge the need to quantify the uncertainty related to the TTD approach in the observational estimates of  $S_{\Delta \rm age}^{\Delta \rm AOU}$ . Following the reviewer's suggestion, we will apply the TTD approach to the outputs of NorESM2-LM to estimate the mean age and derive  $S_{\Delta \rm age}^{\Delta \rm AOU}$ . By comparing the results with  $S_{\Delta \rm age}^{\Delta \rm AOU}$  derived from the ideal age, we will quantify the uncertainty in the observational estimate related to the TTD approach. As commonly done (e.g., He et al. (2018), Guo et al. (2025)), we will conduct this analysis for only one model. In addition, we will mention the uncertainty related to using a single-tracer, we thanks the reviewer for pointing it out.

- RC: Second, the authors calculated trends of AOU and age in the TS bins. In this case, the trend of age and AOU contains several pieces of information: (i) the mean state of water parcel ventilation timescales and AOU may differ across various geographical locations within the same temperature-salinity (TS) bin; (ii) the actual temporal change of age and AOU over time. Based on your Figures 2 and S5, where many data points indicate age trends over roughly 10 yr yr-1, I suspect that the first factor—the spatial variability within TS bins—is the primary contributor to the observed age trend. Given this, I wonder if it is appropriate to describe this as a "trend" with units of years per year (yr yr-1). Although it seems it does not affect the meaning of  $S_{\Delta \rm age}^{\Delta \rm AOU}$ , which is somehow like the respiration rate, it would be helpful to clarify this point.
- AR: Even if there was no geographical constraint, we believe that thanks to their fine granularity (see supplementary Fig. S5) the TS-bins sample waters that can be roughly considered "the same". It is common to calculate AOU trends (or OUR) following isopycnals or to average over large water mass definitions, here we go a step further and look at changes for very small density classes (each TS-bin), which we believe increases the robustness of the results with respect to those previous approaches. Additionally, we have decided to implement a geographical constraint. We will try values between 160km and 500km based on the horizontal decorrelation length scale for deep ocean temperature and the recommended deployment of deep Argo float (Zilberman et al. , 2023; Johnson et al. , 2015). Below these horizontal length scales it can be considered that the water mass has the same properties. We will clarify the manuscript, indicating this new geographical constraint and updating the  $S_{\Delta age}^{\Delta AOU}$  estimation.
- RC: The title of this study is "Earth system models overestimate the sensitivity of apparent oxygen utilization to age change in the deep ocean," but I find that there is limited discussion and analysis of this specific point in the main manuscript.
- AR: Both reviewers expressed concerns about this result, pointing to potential uncertainties in the observation-based estimation of the sensitivity. We will expand the discussion on this point with the new analysis outlined in our replies to the reviewers' comments. We will also expand the discussion on the implications of this overestimation for current projections of the BCP based on AOU and include that hence the slowing down of circulation might not increase the remineralised carbon stock as much as projected (Weijer et al. , 2022). This leads to a weaker biologically-induced decline in atmospheric CO2 (Frenger et al. , 2024), thereby reducing the compensation for the atmospheric CO2 increase caused by anthropogenic emissions. We will also emphasize our lack of understanding of biological processes, particularly those driving interior respiration (Henson et al. , 2024), which consequently cannot be adequately implemented into models.
- RC: If I understand correctly, the models show a greater increase in AOU per unit water age increase compared

to observations. However, this seems to contrast with findings from previous studies, such as Oschlies et al. (2018), which suggest that models tend to systematically underestimate the observed rates of ocean deoxygenation—mainly driven by increases in AOU—particularly in the deep ocean below 1000 m. It would be helpful to add a paragraph discussing this potential discrepancy in the Discussion section.

AR: Thanks for pointing that out. We do not believe it necessarily contrasts with findings from previous studies. Our results indicates indeed that models overestimates the sensitivity of AOU to changes in circulations. However, if the modelled changes in circulation are too weak, AOU changes and subsequent deoxygenation will also be too weak. We will discuss it in the paragraph addressing the importance of changes in circulation (lines 291-302).

**1.2. Minor comments**

- RC: Line 56 58. The oxygen and age are hardly affected similarly by transport in the real ocean due to the spatial heterogeneity of respiration on isopycnals, as found for idealized isopycnals with prescribed patterns of respiration (Koeve et al., 2016; Guo et al., 2023). "Sufficient oxygen concentration" is not the precondition that AOU and age are linear; instead, the even distribution of respiration rate across spatial is.
- AR: Indeed, thanks for pointing that mistake out. We will revise the sentence as follows:

In regions dominated by advection or with an even spatial distribution of the respiration rate, the relation between AOU and age is linear when both are affected similarly by transport (Koeve et al., 2016).

- RC: Line 78 87. I suggest including a table that summarizes key information about the models used in the study.
- AR: Thanks for the suggestion, we will add a table summarizing the following informations for the eleven ESMs considered: which of the considered variables are available as model output in the ESGF database, ocean and biogeochemical model components and reference papers.
- RC: Line 88 89. Do you mean the model outputs are resampled analogously to the observations?
- AR: No, we meant that we only used data points for which a measurement of each variables temperature, salinity, phosphate, oxygen as well as estimates of AOU and age were available. We will adjust the text to be more specific.
- RC: Line 90. It should be GLODAPv2.2023.
- AR: Indeed, thanks for pointing that out. We will revise our manuscript accordingly.
- RC: Line 129. Could you provide a reason why you exclude the Arctic Ocean?
- AR: We separated the Arctic Ocean from the Atlantic Ocean because the linear relationship between AOU trends and age trends was very clearly different. We will indicate that in the revised manuscript.
- RC: Line 131. To clarify, please add a sentence indicating that all longitudes are considered for Southern Ocean water masses.
- AR: We will include this information in the revised manuscript.
- RC: Line 151. What is the difference between B and  $\varepsilon$ ?

- AR: B is the intercept of the linear relationship. It is an output of the linear regression. B is the spatially averaged AOU change when there is no change in age.  $\varepsilon$  is the error of the linear regression in each point. So in each point  $B + \varepsilon$  is the change in AOU that is not linearly related to changes in age. We will clarify this as follows:
  - The equation will be revise to show that B is independent of the grid point while  $\varepsilon$  is not. The new equation will read:

In this work, we intend to express the trends in AOU  $(\frac{dAOU}{dt})$  via trends in age  $(\frac{dage}{dt})$ , in each grid-point X, as follows:

$$\frac{\mathrm{dAOU}}{\mathrm{dt}}(X) = S_{\Delta \mathrm{age}}^{\Delta \mathrm{AOU}} \times \frac{\mathrm{dage}}{\mathrm{dt}}(X) + B + \varepsilon(X) \tag{1}$$

• The aforementioned clarification of B and  $\varepsilon$  will be included as follow:

The slope of the linear regression is the sensitivity of AOU changes to age changes  $(S_{\Delta \rm age}^{\Delta \rm AOU})$ . The intercept of the linear regression, B, represent a spatial average of the changes in AOU when there is no change in age.  $\varepsilon(X)$  is the error of the linear regression in each grid-point. All together,  $B+\varepsilon$  represents the changes in AOU that is not linearly related to changes in age such as changes in remineralization rates.

- RC: Line 161. Jenkins et al. (2024) and Guo et al. (2023) could be included in the citations.
- AR: Indeed, they will be included in our revised manuscript.
- RC: Line 171-180. See my major concern 1.
- AR: See reply to major concern 1.
- RC: Line 195 267. I suggest dividing the Results section into several subsections for improved clarity and organization.
- AR: We considered that when writing the manuscript and choose not to, because the results section in its current form is not very long. However, we will follow the advice from the reviewer and add this organisation during the revision.
- RC: Line 219 221. AOU versus age across water masses can be the reason, but also the change of local biogeochemical processes?
- AR: Indeed. We will revise these lines to be more specific about the reasons for the break down of the linear relationship within the water masses, including 1) significant contribution of mixing over advection, 2) spatial variability of respiration and 3) local changes in biogeochemical processes.
- RC: Figure 2. Are the blue dots representing only values with a significant trend? If so, please clarify this. Otherwise, I am curious why Southern Ocean and Atlantic light/dense water make up around 98% of the deep ocean, yet many points remain grey—are these from the upper 1000 m?
- AR: In Figure 2, the blue shading shows only the values with significant trends belonging to the four water-masses used in the work. The grey background shows the entire ocean, including the non-significant trends and the surface ocean. We will clarify that in the caption of the figure as follows:

The blue shading shows the number of data point for each bin of age trends and AOU trends for the Southern and Atlantic light/dense waters, accounting only for point where age and AOU trends are significant.

The grey area does not indicate the number of points but rather the range of age and AOU trends in the entire ocean. Typically, trends in the upper 1000 meters are stronger than those in deeper regions, thus covering a larger portion of the figure.

- RC: Figure 4. Please improve the visualization of panel (a), as it currently appears quite cluttered, despite its importance. Are the zonal section panels essential, or could they be simplified or omitted to enhance clarity?
- AR: As the zonal section are extensively discussed in the results section, we choose to keep them. To enhance the clarity of the figure, we will split this figure in two: One figure containing only the bar plot and one containing only the sections. The current panel (a) of the figure will be simplified by removing information not use in the manuscript such as the trends in AOU for the global and deep ocean.
- RC: Line 279 290. This section is overall too descriptive. Additionally, as I mentioned in Major concern 1, I am concerned about potential inconsistencies in the methodology when estimating  $S_{\Delta \rm age}^{\Delta {
  m AOU}}$  in models versus observations.
- AR: The aim of this paragraph is to discuss the overestimation of  $S_{\Delta age}^{\Delta AOU}$  by the models. We will extend this paragraph to include the results from the extra analysis outlined in our reply to the reviewers' first comment.
- RC: Line 299 302. Guo et al. (2023) could be cited here. These authors proposed two possible explanations for weak connection between age change and AOU change. First, changes in ocean circulation within a warming climate could alter water mass composition, as different water masses with varying biogeochemical histories are recombined over time—potentially introducing younger yet more oxygen-depleted waters into a given region. Second, local biological activity may influence the AOU signal independently of ventilation. For instance, even if ventilation slightly increases, an increase in local respiration rates could cause AOU to rise, reflecting biological consumption rather than changes in physical mixing.
- AR: Thanks for pointing to this paper. In our revision, we will cite this paper to illustrate how the linear relationship between the trends in AOU and age break down in a transient climate.

**2. Reviewer #2**

RC: This study investigates the relationship between apparent oxygen utilisation (AOU) trends and water mass age trends in the deep ocean using Earth system models and observational data from GLODAPv2. The authors establish linear relationships between these variables to assess the biological carbon pump's sensitivity to circulation changes under contemporary and future climate scenarios. While this work addresses an important aspect of ocean biogeochemistry relevant to climate projections and would be of interest to the journal, several aspects require further clarification or investigation to enhance the paper's overall strength.

**2.1. Major comments**

- RC: Water Mass Definition Methodology: The water mass classification using PO\* thresholds appears somewhat arbitrary and model-dependent. The authors mention "iterative testing" to refine thresholds but provide insufficient detail about this process. Since most of the results are based on this classification, I believe a more detailed section about the derivation of these thresholds is necessary. I also don't understand why there is a difference between the number of water mass definitions for ESMs and observations (four vs two water masses).
- AR: These values were based on zonal averages of the PO-tracer in the Atlantic, Pacific and Indian Ocean. We chose these thresholds with the aim of separating water masses coming from the Southern Ocean and from the North Atlantic as recommended in Broecker et al. (1998). Since biogeochemical properties of water masses are different between models and observation we made some choices with the goal of separating the origin of the water mass while selecting the majority of the deep ocean. To make our approach more objective, we will define the threshold based on the PO-tracer values in the upper 500 meters of the Southern Ocean and subpolar North Atlantic for each model and describe this in our revised manuscript. We do not expect this choice to significantly impact our results.

Concerning the number of water masses, we divided the water masses define from the PO-tracer into two density class because this division improved the correlation between trends in AOU and age. The identifycation of relationships between AOU and age within density classes is commonly done (e.g. Sulpis et al. (2023)). Doing the same for the observation was not possible because it would further reduce the number of data point, making it too low to identify a relationship between trends in AOU and age. We will clarify these choices in our revised manuscript.

- RC: Linear Relationship Assumption Validity: While the authors acknowledge in the introduction that linear relationships between AOU and age can break down in certain regions and under specific conditions of the BCP. It would be interesting to further discuss under which conditions this linearity assumption fails in a transient climate scenario. For example, it is unclear to me under which conditions this linear relation still applies (regionally or globally) on SSP5-8.5.
- AR: We briefly addressed this when discussing potential reasons why the linear relationship between AOU trends and age trends is less robust in the lighter water, mentioning changes in export production or more spatially heterogeneous respiration rates. We agree that it would be valuable to further discuss this aspect. The condition causing a break down of the linear relation ship between AOU and age strong mixing over advection and spatially heterogeneous respiration also apply to the trends. These conditions may change over time in a transient climate, being satisfied for certain period but not for others. In addition, changes in circulation can recombine water masses differently bringing in waters with varying histories; waters can follow different pathways and go through different respiration fields (Guo et al., 2023). Temporal changes in biogeochemistry, such as respiration rates, could also disrupt the linear relationship. We will include the above mentioned points in the revised manuscript.
- RC: Drift Correction Methodology: The drift correction procedure using a piControl simulation makes sense, however, it is not specified how the actual years of the piControl are adjusted to fit 1850-2099.
- AR: The piControl years have been chosen based on the information available in the netcdf files indicated which year of the piControl was used to start the historical simulations. We will clarify this in the revised manuscript.
- RC: Observational Data Limitations and T-S Binning: The T-S space binning approach for observational data analysis is not well justified. The choice of bin resolution (0.027 and 0.0023) seems arbitrary, and the methodology results in very small sample sizes (only 23 and 118 significant joint trends for Southern and

Atlantic water masses). The authors should better emphasise the reason behind using this TS binning.

- AR: The TS-bin resolution results from the trade off between 1) computing significant trends in age and AOU in each TS-bin and 2) getting enough trend estimates to be able to identify a significant relationship between AOU trends and age trends. A finer TS-bin resolution would increase the number of trends estimated but will reduce the number of data in each TS-bin and thus reduce the significance of each trend. On the other hand, a coarser TS-bin resolution may increase the significance of each trend but will reduce the number of trends estimated. To clarify these points, we will revise our supplement to show the impact of choosing different TS-bin sizes on the sample size and the estimation of  $S_{\Delta age}^{\Delta AOU}$  and clarify this point in the revised manuscript as well.
- RC: On the significance of the observed overestimation in ESMs: The authors show clearly that observational values are "on the low side of the ESMs range" which indicates overestimation of the sensitivity of AOU to age change. While they briefly acknowledge limitations in computing estimates for the contemporary period based on observations (lines around 279), I'm not totally convinced that the overestimation in ESMs is not due to having more data. For instance, if the ESMs were sampled at the observational locations and the analysis repeated with the same methodological choices as for observations data (e.g. same water mass definition, same trend calculation approach with the same TS binning), are we going to still going to observe an overestimation?
- AR: We agree with the reviewer but we would rather question the uncertainty in the observation estimate due to the lack of data and the methodological choices. To address this, we will test the methodological approach using the model outputs from NorESM2-LM. We will compute the trends in AOU and age in the TS space, and then derive  $S_{\Delta \rm age}^{\Delta \rm AOU}$ . The comparison with the analysis in the geographical space will measure the uncertainty. Comparing this with the analysis in geographical space will measure the uncertainty. Additionally, we will subsample model outputs to further assess the uncertainty. The revised manuscript will report and discuss the new uncertainty estimate, including that related to the TTD methods mentioned by the other reviewer.
- RC: I hope that the authors will understand my comments in a constructive way, and that I value their work and the time they invested in the preparation of the manuscript. It might be that I have misunderstood something, in this case, if something wasn't clear for me as a reviewer, it is possible that it wouldn't be clear also for the readers.
- AR: We appreciate the constructive comments and thank the reviewer for their feedback, which is valuable to strengthen our manuscript.

**References**

- Guo, H., Kriest, I., Oschlies, A., & Koeve, W. (2023). Can Oxygen Utilization Rate Be Used to Track the Long-Term Changes of Aerobic Respiration in the Mesopelagic Atlantic Ocean?. Geophysical Research Letters, 50(13), e2022GL102645.
- Guo, H., Koeve, W., Oschlies, A., He, Y. C., Kemena, T. P., Gerke, L., & Kriest, I. (2025). Dual-tracer constraints on the inverse Gaussian transit time distribution improve the estimation of water mass ages and their temporal trends in the tropical thermocline. Ocean Science, 21(3), 1167-1182.
- Jenkins, W. J. (1982). Oxygen utilization rates in North Atlantic subtropical gyre and primary production in oligotrophic systems. Nature, 300(5889), 246-248.

- Koeve, W., & Kähler, P. (2016). Oxygen utilization rate (OUR) underestimates ocean respiration: A model study. Global Biogeochemical Cycles, 30(8), 1166-1182.
- Oschlies, A., Brandt, P., Stramma, L., & Schmidtko, S. (2018). Drivers and mechanisms of ocean deoxygenation. Nature geoscience, 11(7), 467-473.
- Sulpis, O., Jeansson, E., Dinauer, A., Lauvset, S. K., & Middelburg, J. J. (2021). Calcium carbonate dissolution patterns in the ocean. Nature Geoscience, 14(6), 423–428. https://doi.org/10.1038/s41561-021-00743-y
- He, Y., Tjiputra, J., Langehaug, H. R., Jeansson, E., Gao, Y., Schwinger, J., & Olsen, A. (2018). A Model-Based Evaluation of the Inverse Gaussian Transit-Time Distribution Method for Inferring Anthropogenic Carbon Storage in the Ocean. Journal of Geophysical Research: Oceans, 123(3), 1777–1800. https://doi.org/10.1002/2017jc013504
- Henson, S., Baker, C. A., Halloran, P., McQuatters-Gollop, A., Painter, S., Planchat, A., & Tagliabue, A. (2024). Knowledge Gaps in Quantifying the Climate Change Response of Biological Storage of Carbon in the Ocean. Earth's Future, 12(6), e2023EF004375. https://doi.org/10.1029/2023EF004375
- Weijer, W., Cheng, W., Garuba, O. A., Hu, A., & Nadiga, B. T. (2020). CMIP6 Models Predict Significant 21st Century Decline of the Atlantic Meridional Overturning Circulation. Geophysical Research Letters, 47(12), e2019GL086075. https://doi.org/10.1029/2019GL086075
- Sulpis, O., Trossman, D. S., Holzer, M., Jeansson, E., Lauvset, S. K., & Middelburg, J. J. (2023). Respiration Patterns in the Dark Ocean. Global Biogeochemical Cycles, 37(8), e2023GB007747. https://doi.org/10.1029/2023GB007747
- Frenger, I., Landolfi, A., Kvale, K., Somes, C. J., Oschlies, A., Yao, W., & Koeve, W. (2024). Misconceptions of the marine biological carbon pump in a changing climate: Thinking outside the "export" box. Global Change Biology, 30(1), e17124. https://doi.org/10.1111/gcb.17124
- Johnson, G. C., Lyman, J. M., & Purkey, S. G. (2015). Informing Deep Argo Array Design Using Argo and Full-Depth Hydrographic Section Data. Journal of Atmospheric and Oceanic Technology, 32(11), 2187–2198. https://doi.org/10.1175/JTECH-D-15-0139.1
- Zilberman, N. V., Thierry, V., King, B., Alford, M., André, X., Balem, K., et al. (2023). Observing the full ocean volume using Deep Argo floats. Frontiers in Marine Science, 10. https://doi.org/10.3389/fmars.2023.1287867
- Broecker, W. S., Peacock, S. L., Walker, S., Weiss, R., Fahrbach, E., Schroeder, M., et al. (1998). How much deep water is formed in the Southern Ocean? Journal of Geophysical Research: Oceans, 103(C8), 15833–15843. https://doi.org/10.1029/98JC00248